# Beyond Classification: Taxonomic Discovery of Novel Categories in Open-World Settings

## Abstract

Generalized Category Discovery (GCD) has emerged as an important open-world learning problem that aims to automatically cluster partially labeled data. While significant progress has been made in GCD, existing methods focus solely on discovering new categories without capturing their semantic meaning that is a crucial capability for real-world applications. To address this limitation, we propose a novel setting called Taxonomic Discovery of Novel Categories in Open-World Setting (Taxo-GCD) that incorporates hierarchical classification into the GCD framework. Our approach not only identifies novel categories but also determines their taxonomic attributes by learning to recognize concepts at different hierarchical levels. Unlike traditional GCD methods, we develop a parametric classification approach that combines a hierarchical classifier with a unified learner network, further enhanced through a meta-learning architecture. This architecture employs a bi-level optimization scheme with inner update using self-supervised loss for fast test-time adaptation and outer updates for overall optimization. Extensive experiments demonstrate that Taxo-GCD achieves state-of-the-art performance on three standard visual recognition benchmarks, advancing open-world recognition by enabling semantic understanding of novel categories through their taxonomic relationships.

## 1 Introduction

Real-world machine learning systems must operate under open-world conditions, where the assumption that all classes are known at training time rarely holds. In such scenarios, models must not only recognize known categories but also discover and interpret novel ones. Addressing this challenge requires moving beyond standard classification tasks to more flexible and adaptive paradigms. One such paradigm is Novel Class Discovery (NCD) (Han et al., 2019), which focuses on discovering novel categories in unlabeled data. However, in the NCD setting, the labeled and unlabeled classes are strictly disjoint. More recently, Generalized Category Discovery (GCD) (Vaze et al., 2022) has been proposed to address this issue, which extends NCD by allowing unlabeled data to include both known and unknown categories, making it more suitable for real-world scenarios. The goal of GCD is to cluster novel classes while maintaining classification performance on known classes.

However, NCD/GCD still has limitations in real-world scenarios. We can consider a situation, for example, an autonomous vehicle is driving on the street. Suddenly, a "Basset Hound" appears, but the vehicle does not recognize what it is. At this moment, the autonomous system is uncertain about the next decision to make even knowing the front object is a new class. In reality, every object is associated with multiple semantic attributes, e.g., the "Basset Hound" also belongs to the categories "Dog", "Mammal", and "Animal". In this situation, when encountering new classes, such as "basset hound", GCD can only ideally identify this as a new class without providing any additional information. However, this is not how human works. When humans encounter such new object, although we may not know the exact class, we at least know it is a mammal. Knowing this coarser level class can already provide meaningful information and insights into these discovered categories. For the vehicle, recognizing the object as a mammal could immediately prompt a safer decision, such as stopping or changing lanes.

Moreover, categorical definitions evolve dynamically over time, e.g., Black Swans was discovered until in 1967 which fundamentally altered the existing conception of swan species. As shown in

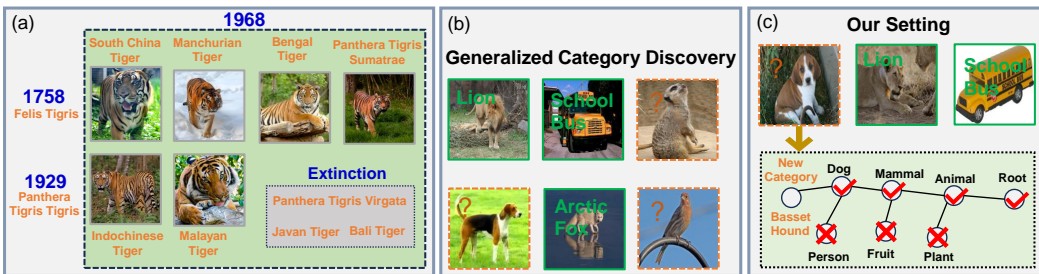

Figure 1: **The taxonomic history of tigers**. (a) In 1758, when Swedish naturalist Carl Linnaeus named the species Felis tigris, with the type locality being Bengal. In 1929, Reginald Pocock revised the name to Panthera tigris tigris. In 1968, Czech biologist Vratislav Mazák proposed a classification of 8 subspecies, including 3 species that are now extinct. Our proposed Taxo-GCD (c) differs from existing Generalized Category Discovery (GCD) approaches (b). While standard GCD focuses solely on identifying novel categories within unlabeled data containing both known and unknown classes, Taxo-GCD significantly extends this capability by additionally inferring the hierarchical attributes and taxonomic relationships of discovered categories.

Fig. 1(a), class attributes may expand or diminish, similar to how tiger classification has transitioned from traditional morphological to modern genetic-based taxonomies. This inherent dynamism means we cannot establish absolute, immutable definitions for objects. Crucially, conventional GCD approaches fail to capture this evolving nature of categories.

To address these limitations, we introduce a novel framework that simultaneously discovers novel categories and identifies their hierarchical attributes. Fig. 1(b&c) illustrates the key distinctions between traditional GCD approaches and our approach. Our method develops representations capable of hierarchical classification, enabling identification of both superclasses and subclasses while dynamically maintaining the class hierarchy. Most importantly, our framework adapts to conceptual changes in categories, allowing for continuous knowledge expansion as taxonomic understanding evolves.

To address this challenging setting, we propose an end-to-end trainable framework that integrates a parametric classifier with a deep model, enhanced via meta-learning. Our method offers two key advantages: (i) effective representation learning for novel category discovery, and (ii) accurate hierarchical classification that preserves taxonomic relationships. Unlike traditional GCD methods (Vaze et al., 2022; Fei et al., 2022) that rely on non-parametric pseudo-labeling and cannot jointly optimize category boundaries, our parametric approach enables unified learning of all categories. Moreover, most existing works adopt a transductive setup, which is less practical for real-world deployment. Inspired by SimGCD's parametric formulation (Wen et al., 2023) and entropy regularization, we further incorporate a meta-learning strategy, Model-Agnostic Meta-Learning (MAML), to improve adaptability. MAML enables nested optimization with an inner update (simulating test-time adaptation) and an outer update for global optimization. During meta-training, we combine supervised contrastive learning on labeled data with self-supervised learning on all data, and introduce self-distillation to strengthen the classifier. Crucially, our method supports test-time adaptation using self-supervised and self-distillation losses on streaming unlabeled samples, addressing a key limitation of existing GCD models that remain fixed at inference. Consequently, the model parameters are optimized to enable fast adaptation during meta-testing.

Furthermore, we integrate taxonomic classification into the meta-learning framework to enable hierarchical categorization. After training, the model performs fast adaptation with unlabeled data, updating its parameters to simultaneously address both tasks. In our setting, we also consider the label shift problem, where an object's attributes may evolve or expand over time. For instance, scientists may one day discover a new species under the category "Basset Hound", and our method can dynamically recognize and assign it to the appropriate class nodes.

To evaluate the effectiveness and generality of our method, we have applied our approach to some existing representative methods. Extensive experiments on CIFAR100-LT (Cao et al., 2019), CIFAR100-

Hier (Wu et al., 2024a) and AWA2-LT (Xian et al., 2019) for image classification show that our work significantly improves performance.

Our contribution can be summarized as follows:

- We propose a new setting, Taxonomic Discovery of Novel Categories in Open-World Settings (Taxo-GCD), which extends beyond traditional GCD by simultaneously: (i) discovering novel categories, and (ii) inferring their taxonomic relationships. This dual capability enables a comprehensive semantic understanding of newly discovered classes within a structured conceptual hierarchy.

- To address this challenge, we develop a parametric optimization framework that combines a hierarchical classifier with a unified learner network, enhanced through Model-Agnostic Meta-Learning (MAML) to improve adaptability to novel categories.

- At test time, our model leverages test-time adaptation (TTA) within the MAML framework to perform fast adaptation on unlabeled samples, enabling effective generalization to unseen categories.

## 2 RELATED WORK

**Generalized Category Discovery (GCD).** The *Generalized Category Discovery (GCD)* task, recently formalized by Vaze et al. (Vaze et al., 2022), extends *Novel Class Discovery* (NCD) (Han et al., 2019; 2021; Zhao & Han, 2021; Zhong et al., 2021; Fini et al., 2021), which assumes unlabeled data contains only novel classes (Han et al., 2019; Fini et al., 2021). GCD relaxes this by allowing unlabeled data to include both known and novel categories, reflecting a more practical but complex setting (Cao et al., 2021; Fini et al., 2021; Oliver et al., 2018). Existing GCD approaches (Wu et al., 2023a; Choi et al., 2024; Cendra et al., 2024; Vaze et al., 2022; Wen et al., 2023; Fei et al., 2022) mainly fall into non-parametric and parametric methods. Non-parametric methods combine supervised and self-supervised learning with clustering (e.g., semi-supervised k-means) (Vaze et al., 2022), but suffer from scalability issues on large datasets. Parametric methods (Vaze et al., 2022; Fei et al., 2022) address this by using trainable classifiers, yet often overfit to known classes. SimGCD (Wen et al., 2023) mitigates this via entropy regularization, achieving notable gains. Nonetheless, both paradigms lack the ability to provide semantic understanding of discovered categories, which is a key limitation our work addresses.

**Taxonomic Classification.** Hierarchical classification organizes classes into taxonomic structures via recursive superclass-subclass relationships, enabling predictions at multiple semantic levels. Wu et al. (Wu et al., 2019) propose a path-based softmax classifier using a "win" metric to aggregate likelihoods along root-to-leaf paths. Bertinetto et al. (Bertinetto et al., 2020) introduce the Hierarchical Cross-Entropy (HXE) loss, combining conditional likelihoods with a soft-label variant of label smoothing (Müller et al., 2019). Deep-RTC (Wu et al., 2020) advance this by using stochastic tree sampling with losses evaluated at random hierarchy cuts. Valmadre (Valmadre, 2022) improve inference with threshold-based rules. More recently, ProTeCt (Wu et al., 2024a) leverage vision-language models to perform effective hierarchical classification under zero- and few-shot settings. In this work, we extend these efforts by integrating taxonomic classification into Generalized Category Discovery (GCD), enabling novel category discovery while preserving hierarchical relationships within an evolving taxonomy.

**Meta-Learning.** The fundamental objective of meta-learning (Santoro et al., 2016; Bateni et al., 2020; Chen et al., 2021; Finn et al., 2017; Wu et al., 2023b; Snell et al., 2017) is to develop models capable of learning how to learn, enabling fast adaptation to new tasks through the acquisition of transferable knowledge and learning strategies. Typical meta-learning methods utilize bi-level optimization to train a model that is applicable for downstream adaptations. Our framework is built upon Model-Agnostic Meta-Learning (MAML) (Finn et al., 2017), employing its bi-level optimization structure consisting of inner loop update for task-specific adaptation and outer loop update for meta-parameter optimization. This architecture enables fast model adaptation through gradient-based inner loop optimization, a paradigm successfully demonstrated in various computer vision applications including test-time adaptation (Zhang et al., 2021; Wu et al., 2024c; Liu et al., 2023; Wu et al., 2024b), which addresses domain shift through dynamic adjustment of pre-trained models using unlabeled test data (Liu et al., 2021; Niu et al., 2023; Bartler et al., 2022; Liu et al.,

2023; Wu et al., 2024b). In this work, the inner loop adaptation operates in an unsupervised manner, facilitating the acquisition of discriminative representations for novel samples.

## 3 PRELIMINARY

In this section, we first describe the problem definition and our setting.

**Problem Definition and Setup.** The goal of Taxo-GCD is to enable an offline-trained model to both discover and perform taxonomic classification of novel object categories using unlabeled data that contains a mixture of known and previously unseen classes. We consider a dataset $\mathcal{D}$ consisting of two subsets: $\mathcal{D}_L = \{(\mathbf{x}_i, y_i)\}_{i=1}^N \subset \mathcal{X} \times \mathcal{Y}_L$ (labeled data), and $\mathcal{D}_U = \{(\mathbf{x}_i, y_i)\}_{i=1}^M \subset \mathcal{X} \times \mathcal{Y}_U$ (unlabeled data). In this setting, each sample is associated with hierarchical labels, where $\mathcal{Y}_L$ and $\mathcal{Y}_U$ include both *internal nodes* (coarse-grained categories), and *leaf nodes* (fine-grained categories), denoted as $\mathcal{Y}_L^f$ and $\mathcal{Y}_U^f$, respectively. Similar to the GCD setting, we assume $\mathcal{Y}_L^f \subset \mathcal{Y}_U^f$. However, unlike traditional GCD, our setting also ensures that $\mathcal{Y}_L$ and $\mathcal{Y}_U$ span the full hierarchy of categories, including all coarse-level classes, structured using WordNet (Miller, 1993). During training, the model has access only to the labeled subset $\mathcal{D}_L$. At test time, it is evaluated on streaming unlabeled data from $\mathcal{D}_U$, where it must not only identify whether a sample belongs to a *novel category $y_{\text{novel}}$* (i.e., a previously unseen leaf node) but also assign it to the appropriate internal node in the taxonomy. This joint objective requires the model to perform both novel category discovery and taxonomic classification within a unified framework.

**Taxonomic Classification Setup.** We formalize our hierarchical classification framework through the following mathematical characterization. Let $\mathcal{Y}_\mathcal{T}$ represent the complete taxonomic class hierarchy containing both leaf-level categories and their parent nodes (superclasses). For hierarchical classification, an instance belonging to class $y$ and belongs to ancestors (superclasses) of $y$; In addition, constrained multi-label classification where valid label sets must form complete root-to-leaf paths in the hierarchy, and mutual exclusivity among sibling categories requiring that an instance can only belong to one branch of the taxonomy at any given level. The classification objective focuses on predicting the most specific correct label $y \in \mathcal{Y}_L^f$ while recognizing that any prediction along the ancestral path $\mathcal{A}(y)$ composes of partially correct classifications.

## 4 OUR APPROACH

**Method Overview.** Fig. 2 shows an overview of Taxo-GCD with the meta-learning framework. While conventional non-parametric approaches suffer from suboptimal decision boundaries due to their inability to jointly optimize across all categories, our parametric solution builds upon (Wen et al., 2023) to enable end-to-end trainable category discovery through meta-learning. The framework simultaneously incorporates a deep taxonomic classifier within this meta-learning paradigm, leveraging MAML's bi-level optimization to achieve both robust category discovery and precise hierarchical classification. This dual mechanism allows the model to rapidly adapt to novel categories while preserving their taxonomic relationships with existing classes.

**Model Architecture.** Fig. 2 (a) illustrates the architecture of our proposed framework. The processing pipeline begins with input images undergoing random augmentations to generate two views. These augmented samples are then processed through a Vision Transformer (ViT-B16) encoder $E(\theta)$, pretrained using DINO (Dosovitskiy et al., 2020). During both training and testing, we maintain a fixed parameter configuration for most transformer blocks, with fine-tuning limited exclusively to the final block. The architecture subsequently diverges into two parallel pathways. The first pathway establishes a direct connection between the encoder and a parametric classifier, followed by a multi-layer perceptron (MLP) projection head. This is a big difference between our method and existing non-parametric GCD methods. The second pathway routes encoder outputs through a deep classifier specifically designed for taxonomic classification.

### 4.1 REPRESENTATION LEARNING

Building upon the GCD framework of (Vaze et al., 2022), we employ a hybrid contrastive learning approach to learn discriminative representations, combining self-supervised (Chen et al., 2020) and

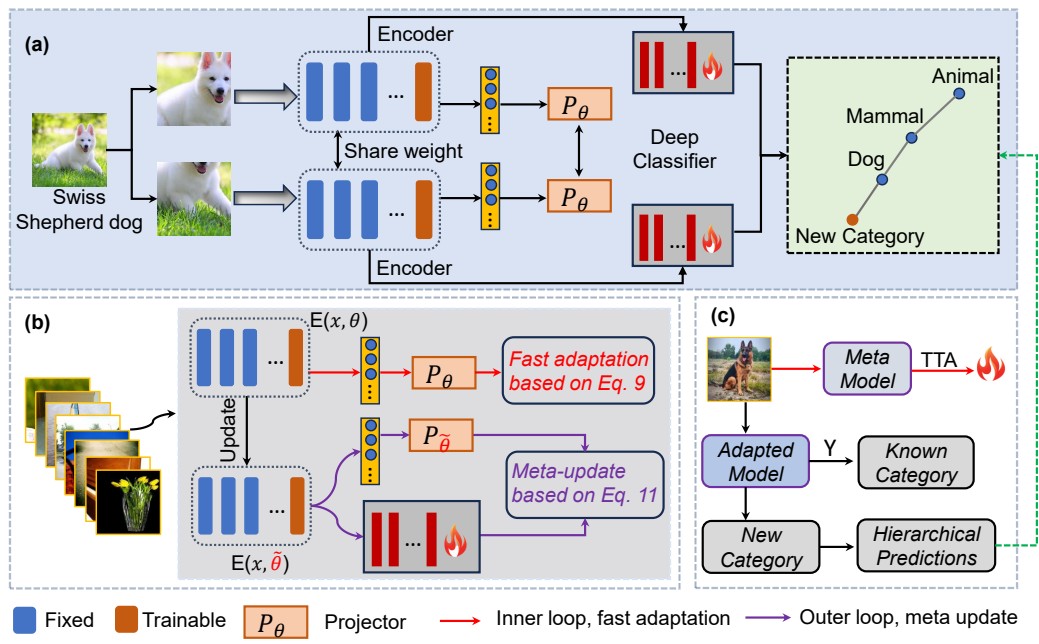

Figure 2: **Conceptually illustration of Taxo-GCD.** (a) shows the overall network of the method. The processing pipeline begins with input images undergoing random augmentations to generate two views. These augmented samples are then processed through a Vision Transformer (ViT-B16) encoder, pretrained using DINO (Dosovitskiy et al., 2020). The first pathway establishes a direct connection between the encoder and a parametric classifier, followed by a multi-layer perceptron (MLP) projection head. he second pathway routes encoder outputs through a deep classifier specifically designed for taxonomic classification. (b) presents training process of the model. During inner loop, the model is optimized by the self-supervised loss with few gradients. During outer loop, the model is updated using supervised loss, and constrain the optimization direction targeted to inner loop. (c) shows the meta-testing stage. Upon encountering new samples, the model undergoes fast adaptation via supervised loss. This adapted model simultaneously achieves two objectives, i.e., novel category discovery and hierarchical classification.

supervised (Khosla et al., 2020) contrastive learning for labeled data while using only self-supervised contrastive learning for unlabeled data. Given two randomly augmented views $x_i$ and $x_i'$ of the same image, the unsupervised contrastive loss is defined as:

$$\mathcal{L}_{rep}^u = -\log \frac{\exp(z_i \cdot z_i'/\tau_u)}{\sum_n \mathbb{I}_{[n\neq i]} \exp(z_i \cdot z_n/\tau_u)}, \tag{1}$$

where $z_i = g(f(x_i))$ represents the $\ell_2$-normalized feature obtained through the backbone feature extractor $f(\cdot)$ and MLP projection head $g(\cdot)$, with $\tau_u$ denoting the temperature value, and $\mathbb{I}_{[n\neq i]}$ is an indicator function. The supervised contrastive loss maintains a similar structure to its unsupervised counterpart. This loss is formally expressed as:

$$\mathcal{L}_{rep}^s = -\frac{1}{|\mathcal{N}(i)|} \sum_{q \in \mathcal{N}(i)} \log \frac{\exp(z_i \cdot z_q/\tau_s)}{\sum_{n=1}^N \mathbb{I}_{[n\neq i]} \exp(z_i \cdot z_n/\tau_s)}, \tag{2}$$

where $\mathcal{N}_i$ indexes all other images in the same batch that hold the same label as $x_i$, and $\tau_s$ denotes the temperature value.

## 4.2 PARAMETRIC STRATEGY FOR GCD

In our architecture, we use the parametric classification paradigm follows the self-distillation (Caron et al., 2021) strategy. Formally, with $K = |\mathcal{Y}_l \cup \mathcal{Y}_u|$ denoting the total number of categories (e.g., CIFAR100 (Krizhevsky et al., 2009) contains 100 categories), we use $C = \{c_1, \ldots, c_K\}$ to stand for

all classes. The classification objectives are then simply cross-entropy loss between the predictions and pseudo-labels:

$$\mathcal{L}_{\mathrm{cls}}^{u} = -\frac{1}{|B|} \sum_{i \in B} \sum_{k} \boldsymbol{q}'^{(k)} \log \boldsymbol{p}^{(k)} - \varepsilon H(\overline{\boldsymbol{p}}), \tag{3}$$

Similarly, The classification objectives between the predictions and ground-truth labels is below:

$$\mathcal{L}_{\mathrm{cls}}^{s} = -\frac{1}{|B^l|} \sum_{i \in B^l} \boldsymbol{y}^{(k)} \log \boldsymbol{p}^{(k)}, \tag{4}$$

During training, we compute soft labels $\boldsymbol{q}'_i$ for each augmented view $\boldsymbol{x}_i$ by applying softmax to the cosine similarities between the latent representation $\boldsymbol{h}_i = f(\boldsymbol{x}_i)$ and the learned prototypes $C$, where $\boldsymbol{y}_i$ represents the one-hot ground truth label. The framework incorporates a mean-entropy maximization regularizer (Assran et al., 2022) to enhance the unsupervised learning objective, where $\overline{\boldsymbol{p}} = \frac{1}{2|B|} \sum_{i \in B} (\boldsymbol{p}_i + \boldsymbol{p}'_i)$ defines the batch-averaged class probabilities and $H(\overline{\boldsymbol{p}}) = -\sum_k \overline{\boldsymbol{p}}^{(k)} \log \overline{\boldsymbol{p}}^{(k)}$ quantifies the prediction entropy.

### 4.3 PARAMETER-SHARING STRATEGY FOR TAXONOMIC CLASSIFICATION

In this work, beyond discovering novel categories, we further identify and assign the corresponding multi-level hierarchical nodes for each new class within the taxonomic structure. We follow Deep-RTC (Wu et al., 2020), utilizing a deep taxonomic classifier to facilitate hierarchical predictions. We employ a parameter-sharing strategy to uncover the relationships between subclasses and their corresponding ancestor nodes. The predictors $f_{\mathcal{Y}_p}$ for all possible label sets $\mathcal{Y}_p$ that can be derived from taxonomy tree $\mathcal{T}$. We use a dynamic CNN architecture with parameter sharing mechanisms. A popular implementation of this constraint, denoted parameter inheritance (PI), reuses parameters of ancestors nodes $\mathcal{A}(n)$ in the predictor of node $n$. The column vector $\mathbf{w}_n$ of $\mathbf{W}_{\mathcal{Y}}$ is then defined as

$$\mathbf{w}_n = \theta_n + \sum_{p \in \mathcal{A}(n)} \theta_p , \quad \forall n \in \mathcal{Y} \tag{5}$$

where $\theta_n$ are non-hierarchical node parameters. Internal node decisions can thus be calibrated by noting that sample $\mathbf{x}_i$ provides supervision for all node-conditional classifiers in its ground-truth ancestor path $\mathcal{A}(y_i)$. This allows the definition of a node-conditional consistency loss per node $n$ of the form:

$$\mathcal{L}_{sis} = \frac{1}{M} \sum_{i=1}^{M} \frac{1}{|\mathcal{A}(y_i)|} \sum_{n \in \mathcal{A}(y_i)} L_{\mathcal{Y}_n}(\mathbf{x}_i, y_{n,i}), \tag{6}$$

where $L_{\mathcal{Y}_n}$ is the loss for the label set $\mathcal{Y}_n$ and $y_{n,i}$ the label of $\mathbf{x}_i$ for the decision at node $n$. Our architecture employs a dynamic CNN network featuring a multi-level hierarchical classifier, where the number of classifier branches corresponds directly to the depth of the taxonomic hierarchy. For a taxonomy with $L$ hierarchical levels (e.g., $L = 5$), we implement $L$ parallel classifier heads.

Building on prior work (Sun et al., 2013; Wu et al., 2020), we employ a parameter sharing strategy where the standardized score of each node is derived from the sum of its ancestor scores. Inference is carried out via greedy top-down traversal. To ensure that internal nodes receive sufficiently high scores even when all ground-truth labels lie at the leaves, we introduce a stochastic tree sampling strategy during training. Specifically, random cuts are generated by sampling a Bernoulli random variable, and the expected cross-entropy is computed:

$$\mathcal{L}_{rand} = \frac{1}{M} \sum_{i=1}^{M} L_{\mathcal{Y}_c}(\mathbf{x}_i, y_i). \tag{7}$$

By considering different cuts at different iterations, the learning algorithm forces the hierarchical classifier to produce well calibrated decisions for all label sets. Due to the differences between hierarchical and flat classifications, we introduce *confidence threshold* (please see supplementary material).

### 4.4 MODEL LEARNING

**Meta-Training.** For each sampled sequence $\mathcal{D}$ from $\mathcal{D}_L$, we allow the model to explore the incoming unlabeled data in an unsupervised manner. To prevent overfitting to seen categories

during knowledge acquisition, we employ bi-level optimization (Finn et al., 2017), taking GCD and taxonomic classification as the meta-objective. The meta-learning procedure is outlined in Alg. 1 of the supplementary material and Fig. 2 (b). Specifically, we decouple the network into $\theta = \{\theta^E, \theta^P\}$, where $\theta^E$ and $\theta^P$ denote the encoder and projection layers, respectively. At each iteration, we first focus on learning from the current samples. In the inner loop, we adapt the model using an unsupervised loss. The parameters $\theta$ are updated on unlabeled samples $\mathcal{D}_{tr}^i = \{\mathbf{x}_{tr}^i\}$ via a few gradient steps:

$$\tilde{\theta}^{E,P} = \theta^{E,P} - \alpha \nabla_{\theta^{E,P}} \mathcal{L}_{inner}(\mathbf{x}_{tr}^i; \theta), \tag{8}$$

where $\alpha$ is a learning rate, and $\mathcal{L}_{inner}(\cdot)$ is the unsupervised loss for the inner loop optimization in MAML. By thoroughly exploring the unlabeled data, the model learns representations suitable for two tasks, i.e., discovering new categories and taxonomic classification. The inner loop loss is defined:

$$\mathcal{L}_{inner} = \gamma_u \mathcal{L}_{rep}^u + (1 - \gamma_u) \mathcal{L}_{cls}^u, \tag{9}$$

where $\gamma_u$ is balancing weight. Eq. 8 simulates how the model performs two-task learning on incoming unlabeled data during test-time. Ideally, the adapted parameters $\tilde{\theta}^{E,P}$ should generalize well to both Generalized Category Discovery (GCD) and taxonomic classification tasks on new samples. To achieve this, we define the meta-objective (global model) for the outer loop meta-optimization as follows:

$$\min_{\theta^E, \theta^P, \theta^H} \sum_{(\mathcal{X}, \mathcal{Y}) \in \mathcal{D}_L} \mathcal{L}_{meta}(\mathcal{X}, \mathcal{Y}; \tilde{\theta}^E, \tilde{\theta}^P, \theta^H), \tag{10}$$

where $\theta^H$ denotes the deep classifiers, and $\mathcal{L}_{meta}(\cdot)$ represents the outer loop loss. Note that the optimization is performed over the original parameters $\theta$, even though $\mathcal{L}_{meta}(\cdot)$ depends on the adapted parameters $\tilde{\theta}^E, \tilde{\theta}^P$, and the fixed $\theta^H$. The $\mathcal{L}_{meta}$ is defined as:

$$\mathcal{L}_{meta} = \gamma_s \mathcal{L}_{rep}^s + (1 - \gamma_s) \mathcal{L}_{cls}^u + \gamma_s \mathcal{L}_{rep}^u + (1 - \gamma_s) \mathcal{L}_{cls}^s + \gamma_r \mathcal{L}_{sis} + \gamma_t \mathcal{L}_{rand}, \tag{11}$$

where $\gamma_s$, $\gamma_r$ and $\gamma_t$ are balancing weights. The meta-objective in Eq. 10 is then optimized using stochastic gradient descent (SGD) as:

$$\theta \leftarrow \theta - \beta \nabla_\theta \sum_{(\mathcal{X}, \mathcal{Y}) \in \mathcal{D}_L} \mathcal{L}_{meta}(\mathcal{X}, \mathcal{Y}; \tilde{\theta}^E, \tilde{\theta}^P, \theta^H), \tag{12}$$

where $\beta$ is learning rate. After meta-training, we obtain an initialization of the model $\theta$ which has been specifically trained to discover and learn novel objects, at the same time, predict hierarchical classification from the unlabeled data.

**Meta-Testing (fast adaptation and inference).** As illustrated in Fig. 2 (c), the procedure outlined in Alg. 1 of the material directly aligns with the evaluation protocol. This design makes sure our meta-objective explicitly optimizes the model for its intended evaluation behavior, maximizing performance. While unsupervised learning introduces inherent uncertainties, our approach constrains the model through a fully supervised meta-objective. Concretely, during meta-testing, unlabeled test data arrive in batches in a streaming fashion. The model is adapted using Eq. 8 for fast adaptation. After this adaptation, the meta-optimized parameters $\theta$ acquire the dual capability of both novel class discovery and taxonomic classification, enabled by these incoming batches.

## 5 EXPERIMENT

### 5.1 EXPERIMENTAL SETUP

**Datasets.** To evaluate our method's effectiveness, we conduct comprehensive experiments on three benchmark datasets. The dataset statistics can be found in Tab. 1.

**CIFAR100-LT** (Cao et al., 2019), a long-tailed variant of CIFAR-100 (Krizhevsky et al., 2009), contains 60,000 images with $32 \times 32$ across 100 classes with an imbalance factor of 0.01.

**CIFAR100-Hier** (Wu et al., 2024a) extends CIFAR-100 with hierarchical labels, comprising 100 leaf nodes and 48 internal nodes, presenting a challenging multi-level classification task.

**AWA2-LT** (Xian et al., 2019) comprises 30,475 images spanning 50 animal classes, with hierarchical relationships derived from WordNet (Miller, 1993), resulting in a 7-level taxonomical tree characterized by structural imbalance.

Table 1: **Dataset statistics used in our evaluation.** Note that 'classes' here refer specifically to the leaf nodes in the taxonomic hierarchy.

|  |  | Labelled | | Unlabelled | |
| --- | --- | --- | --- | --- | --- |
| Dataset | Balance | #Image | #Class | #Image | #Class |
| CIFAR100-LT (Cao et al., 2019) | ✗ | 10747 | 80 | 10000 | 100 |
| CIFAR100-Hier (Wu et al., 2024a) | ✓ | 50.0K | 80 | 10.0K | 100 |
| AWA2-LT (Xian et al., 2019) | ✗ | 20.0K | 40 | 30.0K | 50 |

Table 2: **Results on the three dataset for GCD performance.** We report 'All', 'Old' and 'New' class accuracy.

|  | CIFAR100-LT | | | CIFAR100-Hier | | | AWA2-LT | | |
| --- | --- | --- | --- | --- | --- | --- | --- | --- | --- |
| Methods | All | Old | New | All | Old | New | All | Old | New |
| $k$-means (MacQueen, 1967) | 0.30 | 0.33 | 0.19 | 0.52 | 0.52 | 0.50 | 0.41 | 0.43 | 0.30 |
| GCD (Vaze et al., 2022) | 0.48 | 0.52 | 0.30 | 0.73 | 0.76 | 0.67 | 0.50 | 0.56 | 0.39 |
| RS (Han et al., 2020) | 0.31 | 0.38 | 0.12 | 0.58 | 0.78 | 0.19 | 0.39 | 0.42 | 0.22 |
| UNO (Fini et al., 2021) | 0.41 | 0.54 | 0.20 | 0.70 | 0.81 | 0.47 | 0.48 | 0.50 | 0.22 |
| SimGCD (Wen et al., 2023) | 0.55 | 0.60 | 0.38 | 0.80 | 0.81 | 0.75 | 0.70 | 0.73 | 0.59 |
| SelEX (Rastegar et al., 2025) | 0.51 | 0.55 | 0.33 | 0.82 | 0.85 | 0.76 | 0.71 | 0.68 | 0.61 |
| **Taxo-GCD (Ours)** | **0.63** | **0.67** | **0.48** | **0.83** | **0.85** | **0.78** | **0.72** | **0.76** | **0.64** |

**Evaluation Protocol.** For GCD task evaluated specifically at leaf nodes, we employ the standard clustering accuracy (ACC) metric as established in (Vaze et al., 2022), with important implementation details: $\text{ACC} = \max_{p \in \mathcal{P}} \frac{1}{|\mathcal{D}^u|} \sum_{i=1}^{|\mathcal{D}^u|} \mathbb{I}(y_i^* = p(\hat{y}_i))$ where $\mathcal{P}$ represents all possible permutations between predicted and ground truth labels, and $y_i^* \in \mathcal{Y}_{leaf}$ denotes ground truth leaf-node labels. For taxonomic classification evaluation, we introduce a specialized metric called Near Leaf Accuracy ($ACC_{nearleaf}$), which measures prediction correctness at the penultimate (second-to-last) level of the hierarchy. In addition, we use hierarchical accuracy ($ACC_{hier}$) to evaluate the performance of a classifier that allows for rejections.

**Experimental Details.** Following GCD (Vaze et al., 2022), we train all methods using a ViT-B/16 backbone pre-trained with DINO (Caron et al., 2021). We use the output of the `[CLS]` token, which has a dimension of 768, as the image feature representation, and fine-tune only the last block of the backbone. Training is conducted for 200 epochs with a batch size of 256, and the learning rates of inner and outer loops $\alpha$ and $\beta$ are set 0.0001, 0.1, respectively. For the balancing weights $\gamma_u$ and $\gamma_s$ in Eq. 9 and Eq. 11, we set both to 0.35. The weights $\gamma_r$ and $\gamma_t$, which are specific to the taxonomic classification task, are set to 0.1 and 0.2, respectively. For temperature value $\tau_u$ and $\tau_s$, are set to 0.07 and 1.0, respectively. All experiments are conducted on an NVIDIA GeForce RTX 3090 GPU.

### 5.2 COMPARISON WITH STATE-OF-THE-ARTS

We introduce a novel problem setting in this paper, for which no existing methods to directly compare. To demonstrate the effectiveness of our approach, we draw comparisons with methods from two related tasks, i.e., Generalized Category Discovery (GCD) at the leaf-node level and taxonomic classification. First, we compare with state-of-the-art methods in GCD, including GCD (Vaze et al., 2022), SimGCD (Wen et al., 2023), UNO+ (Fini et al., 2021), $k$-means (MacQueen, 1967), SelEX (Rastegar et al., 2025), and RankStats (Han et al., 2020). Next, we evaluate against methods for taxonomic classification, such as flat classification, Parameter Sharing and Deep-RTC (Wu et al., 2020).

In Tab. 2, we report the *All/Old/New* class accuracy on three datasets for all methods. As we can see, the proposed method achieves best results compared to other methods on all three datasets. Specifically, our Taxo-GCD surpasses the most recent method SimGCD by 20.83%, 3.85% and 7.81% on CIFAR100-LT, CIFAR100-Hier and AWA2-LT datasets for the final *New* classes accuracy. Besides, our model outperforms the baseline VanillaGCD (GCD) by 23.81%, 12.05% and 30.56% for the final *All* classes. It is obvious that the baseline performs poorly, especially on the *novel* classes. However, the proposed model has a significant gain in discovering *novel* classes.

Table 3: **Results on the three dataset for hierarchical classification performance.** We compute $ACC_{nearleaf}$ and $ACC_{hier}$.

| Methods | CIFAR100-LT | | CIFAR100-Hier | | AWA2-LT | |
|---|---|---|---|---|---|---|
| | $ACC_{nearleaf}$ | $ACC_{hier}$ | $ACC_{nearleaf}$ | $ACC_{hier}$ | $ACC_{nearleaf}$ | $ACC_{hier}$ |
| Flat softmax | 0.82 | 0.88 | 0.74 | 0.76 | 0.80 | 0.84 |
| PS softmax | 0.88 | 0.90 | 0.78 | 0.80 | 0.85 | 0.88 |
| Deep-RTC (Wu et al., 2020) | 0.95 | 0.98 | 0.81 | 0.85 | 0.90 | 0.92 |
| Ours | **0.99** | **0.99** | **0.85** | **0.88** | **0.92** | **0.94** |

We also compare with three taxonomic classification methods. Tab. 3 shows that our method achieves state-of-the-art results.

## 5.3 Ablation Study

We evaluate the individual effects of method components in this section, All ablative experiments are performed on CIFAR100-LT dataset.

**Effect of Adaptation during Meta-Testing.** To evaluate the impact of adaptation during meta-testing, we present ablation study results in $3^{rd}$ row and $4^{th}$ row of Tab. 4 . Overall, the final results with adaptation consistently outperform the baseline. Specifically, in the GCD task, adaptation leads to significant performance improvements. In addition, we observe that the performance on the 'old' classes achieves substantial improvements. However, for the taxonomic classification task, the gains from adaptation are relatively modest. This indicates that the adaptation process primarily benefits fine-grained class separation, which is critical for GCD, while the hierarchical structure in taxonomic classification may already provide sufficient regularization, limiting further gains.

**Effect of Multi-Task Learning.** Our proposed model addresses two tasks, i.e., GCD and taxonomic classification. We conduct ablation studies to evaluate the model's performance when trained on each task individually versus both tasks jointly. As shown in the $5^{th}$, $6^{th}$ and $7^{th}$ rows of Tab. 4, we observe that incorporating the taxonomic classification task helps improve the performance of the GCD task. However, the taxonomic classification task shows minimal improvement when combined with the GCD task. Note that our we inference the hierarchical tree without leaf node in this study. This asymmetry suggests that taxonomic structure provides useful semantic guidance for discovering novel categories, whereas GCD offers limited feedback for improving coarse-level taxonomic inference. Due to space limitation, please see the supplementary material for more ablation studies.

Table 4: **The ablation study results on the CIFAR100-LT dataset.** We evaluate the effects of TTA strategy and mulit-task learning.

| Methods | CIFAR100-LT | | | | |
|---|---|---|---|---|---|
| | All | Old | New | $ACC_{nearleaf}$ | $ACC_{hier}$ |
| Taxo-GCD (baseline) | 0.58 | 0.62 | 0.46 | 0.99 | 0.98 |
| Taxo-GCD w/ Adaptation (Ours) | **0.63** | **0.67** | **0.48** | **0.99** | **0.99** |
| Taxo-GCD w/o Taxo. | 0.62 | 0.65 | 0.47 | – | – |
| Taxo-GCD w/o GCD | – | – | – | 0.992 | 0.991 |
| **Taxo-GCD (Ours)** | **0.63** | **0.67** | **0.48** | 0.992 | **0.994** |

## 6 Conclusion

In this paper, we presented Taxo-GCD, a new framework for taxonomic discovery of novel categories in open-world settings. By integrating hierarchical classification into the GCD paradigm, our approach goes beyond clustering novel categories to also infer their semantic attributes across multiple levels of abstraction. The proposed unified learner network, optimized within a MAML-based meta-learning framework, enables both accurate discovery and efficient test-time adaptation. Extensive experiments on three benchmark datasets confirmed that Taxo-GCD achieves state-of-the-art performance, advancing open-world recognition toward richer semantic understanding. In future work, we aim to extend Taxo-GCD toward continual learning in dynamic and large-scale real-world environments.

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

## A    APPENDIX

Due to space limitations in the main paper, we are unable to include all the details there.

## B    LLM USAGE

For the LLM, we only use it to polish parts of the writing.

## C    THE OPTIMIZATION OF TAXO-GCD

The optimization procedure of Taxo-GCD can be found in Alg. 1.

---

**Algorithm 1** The optimization procedure of Taxo-GCD

---

**Require:** $\alpha, \beta$: learning rates
  1: Randomly initialize parameters $\theta$
  2: **while** not converged **do**
  3: Sample $\mathcal{D} = \{(\mathcal{D}_{tr}, \mathcal{D}_{te})\}$ from $\mathcal{D}_L$ and $\mathcal{D}_U$, respectively.
  4: Generate augmented views on $\mathcal{D}_{tr}$ and $\mathcal{D}_{te}$
  5:    **for** $i = 1, \cdots, T$ **do**
  ▷ Adapt parameters with unlabeled samples using self-supervised loss
  6:      $\tilde{\theta}^{E,P} = \theta^{E,P} - \alpha \nabla_{\theta^{E,P}} \mathcal{L}_{inner}(\mathbf{x}_{tr}^i; \theta)$
  ▷ Evaluate meta-objective
  7:      Random cuts ← Bernoulli
  8:      $\min_{\theta^E, \theta^P, \theta^H} \sum_{(\mathcal{X},\mathcal{Y}) \in \mathcal{D}_L} \mathcal{L}_{meta}(\mathcal{X}, \mathcal{Y}; \tilde{\theta}^E, \tilde{\theta}^P, \theta^H)$
  9:    **end for**
  ▷ Update model parameters
  10:      $\theta \leftarrow \theta - \beta \nabla_\theta \sum_{(\mathcal{X},\mathcal{Y}) \in \mathcal{D}_L} \mathcal{L}_{meta}(\mathcal{X}, \mathcal{Y}; \tilde{\theta}^E, \tilde{\theta}^P, \theta^H)$
  11: **end while**

---

## D    CONFIDENCE THRESHOLD

In hierarchical classification, the inference strategy must fundamentally differ from standard flat classification where prediction simply selects the class with maximum probability. The leaf-node inference $\xi(p) = \arg\max_{y \in \mathcal{L}} p(y)$ appears as an alternative, both approaches prove inadequate - the former always predicting the uninformative root node, and the latter incapable of selecting meaningful internal nodes. To avoids this limitation, it fails to consider potentially meaningful predictions at internal nodes. To resolve these issues, we introduce *confidence threshold* inference:

$$\xi_\tau(p) = \arg\max_{y \in \mathcal{Y}} I(y) \quad \text{subject to} \quad p(y) > \tau \tag{13}$$

where $I(y)$ represents the information content of node $y$. This approach exhibits crucial theoretical guarantees when $\tau \geq 0.5$: there exists a unique path $\mathcal{A}(\hat{y})$ where each node's predicted probability exceeds $\tau$, and the solution is unique provided $I(y)$ strictly increases along tree edges. We choose the $\tau = 0.5$ case as majority inference for its balanced properties.

## E    ABLATION STUDIES

**Hyperparameter Sensitivity.** There are several hyperparameters in our model, and we conduct sensitivity analyses on $\gamma_u$, $\gamma_s$, $\gamma_r$, and $\gamma_t$. For the balancing weights $\gamma_u$ and $\gamma_s$ in Eq. 10 and Eq. 12, we evaluate how performance varies with different values, as illustrated in Fig. 4(a). We find that a value of 0.35 yields favorable results. Additionally, the trends for $\gamma_r$ and $\gamma_t$ are shown in Fig. 4(b), based on which we set them to 0.1 and 0.2, respectively.

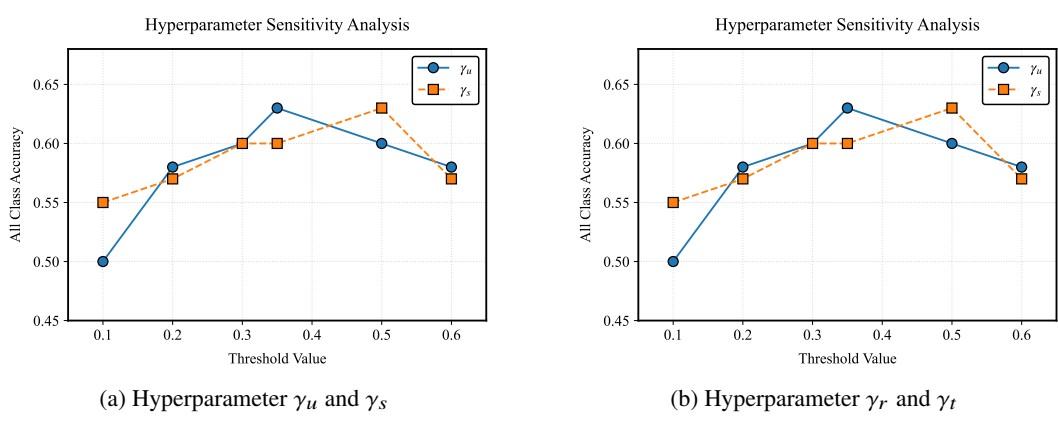

(a) Hyperparameter $\gamma_u$ and $\gamma_s$         (b) Hyperparameter $\gamma_r$ and $\gamma_t$

Figure 3: **The ablation study of hyperparameters.**