# OpenReview forum: "Beyond Classification: Taxonomic Discovery of Novel Categories in Open-World Settings"
_ICLR.cc/2026/Conference — ICLR 2026 Conference Withdrawn Submission_

### Official Review · Reviewer_f9oV · 2025-10-24

**Soundness:** 2
**Presentation:** 2
**Contribution:** 2
**Rating:** 2
**Confidence:** 3

**Summary:**

This paper tackles the task of generalized category discovery - given a labelled training set up some categories learn a model to automatically cluster an unlabelled test set which contains both seen and unseen categories. Particularly this paper focuses on hierarchical categories aiming to both discover the unseen classes in the hierarchy and correctly classify instances of such categories. It achieves this through the proposed method which combines self-supervised learning, taxonomic classification and meta learning. The proposed approach is evaluated on three benchmarks: a long-tailed variant of cifar-100, a hierarchical variant of cifar 100 and long-tailed animal datasets AWA2.

**Strengths:**

- the motivation of bridging taxonomic knowledge and unsupervised category discovery is valid and potentially valuable

- incorporating hierarchical structure into GCD is interesting and can potentially make the results more interpretable

**Weaknesses:**

related work is missing recent GCD works and fails to mention how hierarchies have been used in GCD previously
- recent GCD works such as [A, B, C, D, E, F, G, H] should be added
- the related work should also acknowledge prior work that has considered hierarchies in GCD: SelEX and [A,B,C]
- it does appear these works are different since they consider implicit rather than explicit hierarchies however discussing these and the advantages/disadvantages of an explicit vs. implicit hierarchy would clarify the contribution
***

- unconvincing results
- this is my main issue with the work that can be broken down into several categories below
***

- sota GCD comparison
- the proposed work doesn't improve GCD performance in the main hierarchical dataset tested (Cifar100-Hier - 0.83 vs 0.82 of Selex). It does seem to have more improvement on the other two long-tailed datasets, but this doesn't match the motivation of the proposed method
- related to this, it isn't clear why the main experiments are performed on cifar and AWA2 when there are many fine-grained and hierarchical datasets commonly used in GCD works such as CUB, Stanford Cars, Oxford Pets, FGVC Aircraft and Herbarium
- in relation to the above issue with the related work it isn't clear why the more recent GCD methods [A, B, D, E, F, G, H] also aren't compared to
- only one work from 2024/2025 is compared to SelEX and this is incorrectly cited as 2025 when it should be 2024
***

- ablation
- from the ablation (Table 4) it seems that the joint training of GCD and taxonomic classification doesn't improve results
- the meta-testing does seem to have an impact (Taxo-GCD (baseline) 0.58 vs. Taxo-GCD w/ Adaptation (Ours) 0.63), however the Taxo-GCD baseline performance is already well above the best performing prior work in Table 2, bringing into question why this is and how fair the prior work comparison is
***

- experimental setup
- as well as the datasets chosen (mentioned above) there are several other unclear aspects of the choices made in the experimental setup
- all datasets in GCD setup use a 80:20 split where 80% of the categories are contained in the labelled split. In GCD, particularly on the other datasets mentioned above, a common setup is to have only 50% of the categories in the labelled split increasing the challenge of the task
- it's unclear to me whether the proposed method is benefitting from more information compared to the prior works as the hierarchy gives information about what new classes are expected in inference.
- this paper uses dino features. Typically either dinov2 features or both dino and dinov2 features are used in recent GCD works as dino v2 tends to have better overall performance. The relative improvement tends to be smaller with dinov2 features.
***

[A] Otholt et al. Guided Cluster Aggregation: A Hierarchical Approach to Generalized Category Discovery. WACV 2024
***
[B] Liu et al. Hyperbolic Category Discovery. CVPR 2025.
***
[C] Rastegar et al. Learn to Categorize or Categorize to Learn? Self-Coding for Generalized Category Discovery. NeurIPS 2023.
***
[D] Dai et al. Adaptive Part Learning for Fine-Grained Generalized Category Discovery: A Plug-and-Play Enhancement. CVPR 2025.
***
[E] Xu et al. A Hidden Stumbling Block in Generalized Category Discovery: Distracted Attention. ICCV 2025.
***
[F] Peng et al. MOS: Modeling Object-Scene Associations in Generalized Category Discovery. CVPR 2025.
***
[G] Yang et al. Learning to Distinguish Samples for Generalized Category Discovery. ECCV 2024.
***
[H] Choi et al. Contrastive Mean-Shift Learning for Generalized Category Discovery. CVPR 2024.

**Questions:**

- related work
- The related work section omits discussion of several recent GCD methods (e.g. [A, B, C, D, E, F, G, H]). Could the authors clarify why these works were not included and how their approach compares to them?
- Prior works such as SelEX and [A, B, C] have explored hierarchical structures in GCD. How does the proposed method relate to these approaches?
***

- sota comparison
- The proposed method shows limited improvement over SelEX on the main hierarchical benchmark (CIFAR100-Hier: 0.83 vs. 0.82). Can the authors explain why the gains are modest here despite stronger performance on the long-tailed datasets?
- Why were CIFAR and AWA2 chosen as the main evaluation datasets instead of more standard fine-grained or hierarchical benchmarks (e.g. CUB, Stanford Cars, Oxford Pets, FGVC Aircraft, Herbarium) that are commonly used in recent GCD literature?
- Could the authors clarify why more recent works [A, B, D, E, F, G, H] weren't compared to?
- The baseline Taxo-GCD score (Table 4) is already substantially higher than prior works in Table 2. Could the authors clarify the reason for this discrepancy and whether the comparisons are strictly fair?
***

- ablations
- In Table 4, joint training of GCD and taxonomic classification appears not to improve performance. Could the authors discuss why this might be the case and what the benefit of the proposed method is in light of this?
***

- experimental setup
- The experiments use an 80:20 split (labelled:unlabelled categories), while many GCD works adopt a 50:50 split. Could the authors justify this choice and comment on how it affects the difficulty of the task?
- Does the hierarchical supervision implicitly provide information about the unseen classes during inference, and if so, how is fairness with prior works ensured?
- The experiments use DINO features, while recent GCD works often employ DINOv2 (or both). Could the authors explain why DINOv2 was not considered, and whether they expect the relative performance trends to hold with stronger backbone features?

---

### Official Review · Reviewer_oShb · 2025-10-31

**Soundness:** 2
**Presentation:** 2
**Contribution:** 2
**Rating:** 2
**Confidence:** 5

**Summary:**

This paper introduces Taxonomic Discovery of Novel Categories in Open-World Settings (Taxo-GCD), which extends Generalized Category Discovery by incorporating hierarchical classification through a MAML-based framework. The approach combines parametric classification with deep hierarchical classifiers to simultaneously discover novel categories and infer their taxonomic attributes. However, the technical framework resembles a stack of loss functions and involves numerous hyperparameters. The dataset selection is too limited, lacking experiments on fine-grained datasets with natural taxonomic hierarchies such as CUB-200 and iNaturalist.

**Strengths:**

1．	New problem formulation. The paper introduces Taxo-GCD, which extends Generalized Category Discovery (GCD) to incorporate hierarchical taxonomic classification.
2．	Good results on tested benchmarks (CIFAR100-LT, CIFAR100-Hier and AWA2-LT).

**Weaknesses:**

1. Limited technical novelty; the framework appears to be an integration of SimGCD, Deep-RTC, and MAML.
2. Insufficient completeness of the work, with even noticeable errors. For example, in Eq. (11), supervised and unsupervised weights are set to identical parameters, which contradicts previous GCD experience.
3. Experiments are limited to only three small datasets, lacking testing on fine-grained datasets with natural taxonomic hierarchies such as CUB-200 and iNaturalist.

**Questions:**

See Weaknesses.

---

### Official Review · Reviewer_vEu1 · 2025-11-01

**Soundness:** 2
**Presentation:** 2
**Contribution:** 2
**Rating:** 2
**Confidence:** 5

**Summary:**

The paper proposes a new setting called Taxonomic Generalized Category Discovery (Taxo-GCD): given partially labeled data in an open-world style setup, the model should (i) discover novel categories and (ii) also assign them to the right taxonomic nodes. To do this, the authors build an end-to-end parametric framework, combine SimGCD-style parametric classification with a deep hierarchical classifier, and train it with MAML-style bi-level optimization so that at test time the model can do a few unsupervised adaptation steps on streaming unlabeled data. Experiments on CIFAR100-LT, CIFAR100-Hier, AWA2-LT show better  GCD accuracy than non-hierarchical baselines and better hierarchical accuracy than three traditional taxonomic classification methods.

**Strengths:**

1. **Problem motivation is reasonable**: plain GCD indeed only tells you “this is a new cluster” but not “what kind of thing it is.” Adding hierarchy makes the story closer to real applications.
2. **Technically coherent pipeline**: representation learning (sup+unsup contrastive) → parametric GCD head → hierarchical classifier → MAML outer loop. Every block is at least plausible and grounded in prior work.
3. **Multi-task ablation** shows that injecting the taxonomic task helps GCD a bit, which supports the claim that hierarchy can regularize discovery.

**Weaknesses:**

**W1. Questionable problem setup: “discovery” under a fully known taxonomy**

The paper calls the task taxonomic discovery, but in Sec. 3 it assumes that both the labeled and the unlabeled parts “span the full hierarchy of categories, including all coarse-level classes, structured using WordNet.” That means: the full tree is already known, and the model only has to pick the right internal node for each novel leaf. This is much weaker than “discovering taxonomic relationships.” It is in fact hierarchical classification + GCD on leaves.  In realistic open-world discovery the taxonomy of future unseen classes is not available; often you only have a partial WordNet trim. This assumption should at least be justified and stress-tested (what if a branch is missing? ). Right now, it is effectively an oracle hierarchy.

**W2. Comparison is not strictly fair**

All GCD baselines are non-taxonomic baselines that were not designed to see hierarchical labels at all, but the proposed method does see them (it even trains a deep taxonomic classifier). Yet in Tab. 2 they are compared on the same datasets and the text concludes “our method achieves best results.” This is an information advantage: the proposed model has extra structured supervision; the baselines don't. For a new task, the paper should report a version of SimGCD/UNO/GCD augmented with the same hierarchical loss. Right now, it is hard to tell if gains come from the new setting or simply from more supervision.

**W3. Novelty is combinational**

The three main ingredients are:

1. SimGCD-style parametric GCD with mean-entropy maximization,
2. Deep-RTC hierarchical classifier,
3. MAML-style bi-level optimization for test-time adaptation.
   All three are well-known; the paper's novelty is mostly putting them together in one framework. A “new setting” paper usually needs (i) a very clean formulation and (ii) strong evidence that existing methods fail specifically because of that setting. Here the setting is not so clean (see W1), and the experiments don't prove that “without meta-learning + hierarchy you can't solve it.” So the novelty is on the weak side.

**W4. Missing Key Experiments**

1. Robustness to taxonomy noise / incompleteness — real taxonomies are messy; this paper only studies clean WordNet-style trees.

2. Generalization beyond tiny images — all three benchmarks are small or animals; nothing on, say, ImageNet-100[1], iNat [2], or Bioscan-5M [3] taxonomies.

[1] ImageNet: A large-scale hierarchical image database
[2] The inaturalist species classification and detection dataset
[3] Bioscan-5m: A multimodal dataset for insect biodiversity

**Questions:**

1. Hierarchy availability. During training and testing, do you always provide the full WordNet-style tree (all internal nodes) even for novel leaves? If yes, can you show results when 10–20% of internal nodes are hidden? That would make the “discovery” claim more credible.
2. Meta-testing details. How many inner steps and what batch size do you use for streaming TTA, and what is their influence on model performance? Is it per-image, per-mini-batch, or per-episode?
3. Metric choice. Why introduce Near Leaf Accuracy instead of using common hierarchical metrics (path accuracy / Lowest Common Ancestor / tree-induced F-measure)? Are your improvements still significant there?

---

### Official Review · Reviewer_vgee · 2025-11-02

**Soundness:** 2
**Presentation:** 1
**Contribution:** 2
**Rating:** 4
**Confidence:** 4

**Summary:**

The paper introduces a new problem formulation, Taxo-GCD, which extends the traditional Generalized Category Discovery (GCD) setting to simultaneously discover novel categories and perform hierarchical classification. To address this problem, the authors propose a meta-learning framework that learns a fast adaptation rule on unlabeled data, guided by both the GCD objective and taxonomic classification losses. The method is evaluated on three benchmark datasets with hierarchical label structures and compared against several existing GCD baselines.

**Strengths:**

- The proposed Taxo-GCD problem is interesting and practically meaningful, as it extends the traditional GCD task to incorporate taxonomic relationships among categories.

- The experimental results show competitive or improved performance across three hierarchical benchmarks.

**Weaknesses:**

- Lack of clarity in presentation. The paper’s writing and exposition need substantial improvement. Several key concepts and notations are unclearly defined, making the methodology difficult to follow. For example: 1) Important notations such as y_L^f and y_U^f in Line 175 are never explicitly defined; 2) The section on taxonomic classification (Sec. 4.3) lacks details: the role of the confidence threshold and the definition of the Bernoulli random variable are unspecified.

- Limited technical novelty. The proposed approach essentially combines a MAML-style meta-learning strategy with a hierarchical classification loss. Both components are derived from prior work, and their integration lacks a clear theoretical connection or unified formulation. The contribution therefore appears incremental.

- Inconsistent integration of taxonomy. The method’s incorporation of class hierarchy is not consistent across modules. Sections 4.1 and 4.2 describe GCD learning without considering the taxonomy, while Section 4.3 introduces hierarchical classification without explaining how novel class hierarchies are formed or adapted during meta-learning. This gap raises questions about how the hierarchy is effectively leveraged in the discovery process.

- Questionable assumptions in meta-learning design. The approach assumes that the unlabeled (meta-test) data follow a similar task distribution as the labeled (meta-train) data, which may not hold in real-world GCD scenarios.

- Insufficient experimental validation. 1) The performance gains over strong baselines such as SelEx are marginal on CIFAR100-Hier and AWA2-LT. 2) The ablation study does not analyze the contributions of different loss terms in Equation (11). Given the large number of hyperparameters, it is unclear how they were tuned across datasets. 3) The paper does not include any qualitative or quantitative evaluation of the discovered taxonomic hierarchies, which weakens the empirical support for the main claim.

**Questions:**

- Could the authors clarify how the class hierarchy for novel categories is established during meta-learning?
- How sensitive is the method to hyperparameter tuning, and how were these values selected for each dataset?

---

### Note · Authors · 2025-11-13

I have read and agree with the venue's withdrawal policy on behalf of myself and my co-authors.